# Land Cover Classification from Hyperspectral Images via Local Nearest Neighbor Collaborative Representation with Tikhonov Regularization

**Rongchao Yang** [1,2], **Qingbo Zhou** [1,2,*], **Beilei Fan** [1,2] **and Yuting Wang** [1,2]

1  Agricultural Information Institute, Chinese Academy of Agricultural Sciences, Beijing 100081, China; yangrongchao@caas.cn (R.Y.); fanbeilei@caas.cn (B.F.); wangyuting@caas.cn (Y.W.)
2  Key Laboratory of Agricultural Blockchain Application, Ministry of Agriculture and Rural Affairs, Beijing 100081, China
*  Correspondence: zhouqingbo@caas.cn

**Abstract:** The accurate and timely monitoring of land cover types is of great significance for the scientific planning, rational utilization, effective protection and management of land resources. In recent years, land cover classification based on hyperspectral images and the collaborative representation (CR) model has become a hot topic in the field of remote sensing. However, most of the existing CR models do not consider the problem of sample imbalance, which affects the classification performance of CR models. In addition, the Tikhonov regularization term can improve the classification performance of CR models, but greatly increases the computational complexity of CR models. To address the above problems, a local nearest neighbor (LNN) method is proposed in this paper to select the same number of nearest neighbor samples from each nearest class of the test sample to construct a dictionary. This is then introduced into the original collaborative representation classification (CRC) method and CRC with Tikhonov regularization (CRT) for land cover classification, denoted as LNNCRC and LNNCRT, respectively. To verify the effectiveness of the proposed LNNCRC and LNNCRT methods, the classification performance and running time of the proposed methods are compared with those of six popular CR models on a hyperspectral scene with nine land cover types. The experimental results show that the proposed LNNCRT method achieves the best land cover classification performance, and the proposed LNNCRC and LNNCRT methods not only further exclude the interference of irrelevant training samples and classes, but also effectively eliminate the influence of imbalanced training samples, so as to improve the classification performance of CR models and effectively reduce the computational complexity of CR models.

**Keywords:** land cover classification; hyperspectral images; collaborative representation; sample imbalance; local nearest neighbors

## 1. Introduction

Land cover refers to the biophysical properties of the Earth's surface, which is the most obvious indicator of changes in the Earth's surface [1–3]. The accurate and timely monitoring of land cover types is important for the scientific planning, rational utilization, effective protection and management of land resources [4]. With the rapid development of remote sensing technology, rich remote sensing data sources are provided for the monitoring of land surface information in large areas, such as spaceborne or airborne RGB images [5], multispectral images [6], hyperspectral images [7], and so on. Among them, RGB and multispectral images can provide better spatial features for ground objects. Still, they can only obtain spectral information from several discrete bands, which can easily produce the phenomenon of "same objects with different spectra" and "foreign objects with same spectra", leading to the difficulty of correctly distinguishing between similar land types, such as forest land and cultivated land. For hyperspectral remote sensing images,

each pixel contains hundreds of narrow and continuous spectral bands, which provide abundant spectral and spatial information for the classification of ground objects [8,9]. In view of the advantages of hyperspectral images (HSI), many scholars have applied this technology to the research field of land cover classification [10–12].

Furthermore, most researchers choose statistic-based classification algorithms to establish land cover classification models, such as support vector machines [13], random forests [14], sparse logistic regression [15], and so on. However, such algorithms usually assume that sample data follow normal or multimodal distributions [16] and require a large number of labeled samples for training to fit the models. However, the lack of labeled samples in hyperspectral images can not satisfy the distribution hypothesis of data and will affect the fitting performance of the models.

In the past few years, representation-based classification methods have received great attention. The main idea of these methods is that a test sample is classified through the linear representation of training samples without assuming any data density distribution [17]. Additionally, this kind of algorithm does not need a complex training process, so it avoids the influence of the number of training samples on the model fitting performance. Representation-based classification methods, i.e., sparse representation classification (SRC) [18] and collaborative representation classification (CRC) [19], were originally proposed for face recognition. The essential difference between these two methods is that SRC utilizes $\ell_0$-norm or $\ell_1$-norm regularization to solve the representation coefficient of each test sample, while CRC utilizes $\ell_2$-norm regularization. Reference [19] indicates that CRC is more advantageous than SRC. On the one hand, CRC can obtain a closed-form solution, which makes its computational efficiency higher than SRC. On the other hand, collaborative representation can classify test samples more accurately than sparse representation.

Given the advantages of the collaborative representation (CR) model, it has been widely used in HSI classification in recent years. Li et al. proposed a CR-based classifier that couples nearest-subspace classification (NSC) with a distance-weighted Tikhonov regularization for HSI classification, called nearest regularized subspace (NRS) [20]. In other words, NRS is the Tikhonov regularization version of NSC. Moreover, Li et al. incorporated Tikhonov regularization into the original CRC for HSI classification, denoted as CRT [21]. The main difference between NRS (NSC) and CRT (CRC) is that the former uses training samples of each class to independently represent each test sample (called pre-partitioning). In contrast, the latter uses all training samples from different classes to simultaneously represent each test sample (called post-partitioning). The experimental results in references [20,21] show that both NRS and CRT provide a higher classification accuracy than the original NSC and CRC, which indicates that the Tikhonov regularization can effectively improve the classification performance of CR models. Therefore, the Tikhonov regularization has been applied to many improved collaborative representation classifiers, such as kernel CR with Tikhonov regularization (KCRT) [21], discriminative kernel CR and Tikhonov regularization (DKCRT) [22], structure-aware CR with Tikhonov regularization (SaCRT) [23], and so on. Tikhonov regularization makes the training samples similar to the test samples through large weight vector coefficients by calculating the Euclidean distance between the test sample and all the training samples to improve the classification performance of CR models. However, the Tikhonov regularization term greatly increases the computational complexity of CR models.

Furthermore, the selected dictionary greatly influences the classification performance of CR models [24]. All of the aforementioned CR models collaboratively represent the test sample through the dictionary constructed by all training samples, which may make the training samples that are irrelevant to the test sample produce a negative impact on the weight vector coefficient, resulting in the misclassification of the test sample. To solve this problem, many researchers introduced the nearest neighbor (NN) method into CR models. Li et al. proposed a local within-class CR-based NN (LRNN) method, in which *k*-nearest training samples adjacent to the test sample were selected from each class to

represent the test sample through the way of pre-partitioning [16]. Wei et al. proposed a CRC method based on *k*-nearest neighbors (KNN-CRC) [25]. This method first chooses the *k* nearest neighbors of each test sample from all the training samples, then utilizes these nearest neighbor training samples to collaboratively represent the test sample. On this basis, Yang et al. proposed a multiscale joint CRC method with locally adaptive dictionary (MLJCRC) [26]. The above-mentioned methods construct the dictionary by selecting the training samples that are nearest to the test sample. Different from these methods, Su et al. proposed a *K*-nearest class CRT (KNCCRT) method for HSI classification, in which the training samples from the *K*-nearest classes of each test sample were utilized to construct the dictionary and collaboratively represent the test sample [24]. KNCCRT not only takes advantage of Tikhonov regularization to improve classification performance, but also eliminates the classes that are irrelevant to the test samples to a certain extent, so as to reduce the computational complexity of CR model.

Most of the existing CR models do not consider the imbalance of training samples for each class in the dictionary. In practical land cover classification tasks via hyperspectral remote sensing images, the imbalance of training samples for each class is widespread, which may produce a great impact on the performance of classifiers [27–32]. Some CR models can only passively construct dictionaries using given training samples, such as NRS (NSC), CRT (CRC), KCRT, etc. The more serious the imbalance of training samples is, the more influence they have on the classification performance of these methods. Some CR models construct adaptive dictionaries by excluding training samples or classes that are irrelevant to test samples, such as KNN-CRC, MLJCRC, KNCCRT, and so on. However, these methods do not take the imbalance of training samples into account. The LRNN method considers the imbalance of training samples; the same number of nearest neighbors of test samples are selected from the training samples of each class to construct a dictionary [16]. However, this method utilizes all classes, in which the classes that are irrelevant to the test samples may have an impact on classification performance.

To address the aforementioned problems, a local nearest neighbor (LNN) method is proposed in this paper and introduced into the original CRC and CRT methods for land cover classification, denoted as LNNCRC and LNNCRT, respectively, in which the LNNCRT method is the Tikhonov regularized version of LNNCRC. For the proposed methods, firstly, the same number of nearest neighbors of the test sample is selected from the training samples of each class by Euclidean distance, and a local density measurement method is designed to measure the similarity between the test sample and the selected training samples from each class. Then, the *K*-nearest classes of each test sample are selected by the designed local density to construct a dictionary. Each nearest class is composed of the same number of nearest neighbor samples selected in the first step. Finally, the constructed dictionary represents and classifies the test sample.

The main contributions of this article are as follows:

(1) A local nearest neighbor (LNN) method is proposed and introduced into the original CRC and CRT methods for land cover classification, denoted as LNNCRC and LNNCRT, respectively, which can effectively select the nearest neighbors and nearest classes of each test sample from all the training samples, so as to further exclude the interference of irrelevant samples and classes.

(2) The proposed LNNCRC and LNNCRT methods utilize the same number of nearest neighbors from each nearest class of the test sample to construct dictionary, which can effectively eliminate the influence of imbalanced training samples on classification performance.

(3) Due to the exclusion of the interference of irrelevant samples and classes in a further step, the proposed LNNCRC and LNNCRT methods can not only effectively improve the classification performance of CR models for land cover types, but also reduce the computational complexity of CR models.

## 2. Materials and Methods

### 2.1. Data Collection

In this paper, a land cover classification experiment was carried out using the Pavia University hyperspectral scene provided by the Telecommunication and Remote Sensing Laboratory, Pavia University. This hyperspectral scene was acquired by a Reflective Optics Spectrographic Imaging System (ROSIS) sensor mounted on a flight platform. Additionally, the hyperspectral image contains $610 \times 340$ pixels with a high spatial resolution of 1.3 m, and it provides 103 available spectral bands in the range of 0.43–0.86 μm. Moreover, the hyperspectral scene mainly includes nine land cover types, i.e., *asphalt*, *meadows*, *gravel*, *trees*, *painted metal sheets*, *bare soil*, *bitumen*, *self-blocking bricks*, and *shadows*. the false-color image and ground truth map of the hyperspectral scene are shown in Figure 1.

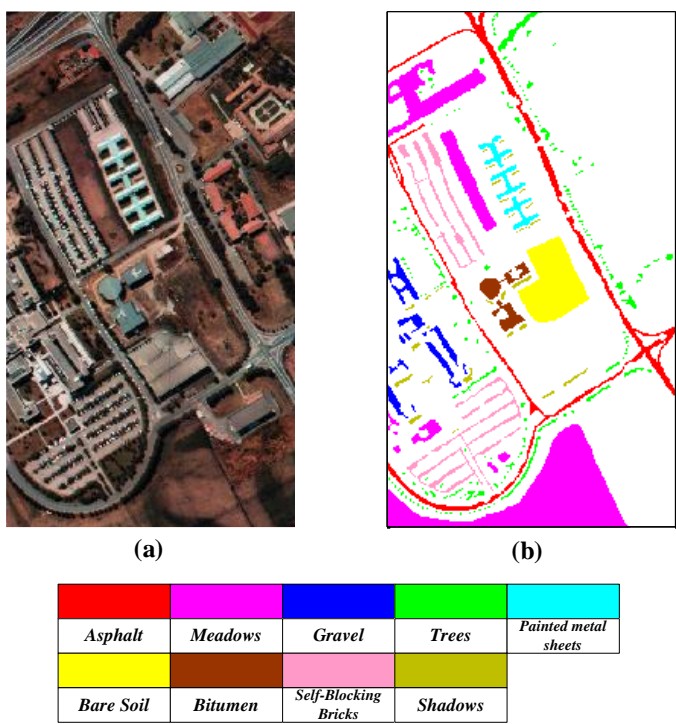

**(a)**        **(b)**

| | | | | |
|---|---|---|---|---|
| *Asphalt* | *Meadows* | *Gravel* | *Trees* | *Painted metal sheets* |
| *Bare Soil* | *Bitumen* | *Self-Blocking Bricks* | *Shadows* | |

**Figure 1.** (**a**) False-color image, and (**b**) ground truth map.

There are 42,776 labeled samples (pixels) in this hyperspectral scene, and the number of samples for each class is extremely unbalanced. To evaluate the classification performance of the proposed methods under the condition of unbalanced training samples, 10% of the labeled samples in each class are randomly picked as the training samples, 20% of the labeled samples in each class are randomly picked as validation samples, and the remaining samples are as test samples, in which the validation samples are utilized to optimize the parameters of the classification models. The specific division of samples is shown in Table 1. Since it is difficult to obtain the real label information of pixels in actual hyperspectral images, this paper attempts to classify a large number of test samples with a small number of training samples. Therefore, the number of training samples selected in this paper is significantly lower than that of test samples.

**Table 1.** The division of samples for each class in the hyperspectral scene.

| No. | Class | Total Samples | Training Samples | Validation Samples | Test Samples |
|---|---|---|---|---|---|
| 1 | *Asphalt* | 6631 | 663 | 1326 | 4642 |
| 2 | *Meadows* | 18,649 | 1865 | 3730 | 13,054 |
| 3 | *Gravel* | 2099 | 210 | 420 | 1469 |
| 4 | *Trees* | 3064 | 306 | 613 | 2145 |
| 5 | *Painted metal sheets* | 1345 | 135 | 269 | 942 |
| 6 | *Bare soil* | 5029 | 503 | 1006 | 3520 |
| 7 | *Bitumen* | 1330 | 133 | 266 | 931 |
| 8 | *Self-blocking bricks* | 3682 | 368 | 736 | 2577 |
| 9 | *Shadows* | 947 | 95 | 189 | 663 |
| | All classes | 42,776 | 4278 | 8555 | 29,943 |

*2.2. Classification Methods*

The KNCCRT method attempts to employ *K*-nearest classes of each test sample to construct the dictionary and linearly represent the test sample [24]. Additionally, the proposed LNNCRC and LNNCRT methods in this paper are inspired by the KNCCRT method, in which LNNCRT is the Tikhonov regularized version of LNNCRC. Different from the KNCCRC and KNCCRT methods, the proposed LNNCRC and LNNCRT methods not only consider the *K*-nearest classes of each test sample, but also take the same number of nearest neighbors of each test sample in the within-class training samples into account, so as to exclude the training samples irrelevant to the test samples to a considerable extent and eliminate the influence of imbalanced training samples on classification performance. Furthermore, a local density measurement method is designed to select the *K*-nearest classes of each test sample instead of simply using Euclidean distance as the measure of similarity. To verify the effectiveness of the proposed LNNCRC method, it is compared with the non-regularized version of KNCCRT (i.e., KNCCRC). In this section, the principles of the KNCCRC, KNCCRT, LNNCRC, and LNNCRT methods are introduced, respectively.

Suppose that $\mathbf{X} = [x_1, x_2, \cdots, x_N] \in \mathbb{R}^{B \times N}$ represents a HSI land cover scene with $C$ classes and $N$ labeled training samples, where $B$ represents the number of hyperspectral bands. Additionally, $\mathbf{X}_l = [x_{l,1}, x_{l,2}, \cdots, x_{l,N_l}]$ represents the training sample set of the $l$th class, $l \in \{1, 2, \cdots, C\}$, where $N_l$ is the number of the training samples in the $l$th class, and $\sum_{l=1}^{C} N_l = N$. Consequently, $\mathbf{X}$ can also be denoted as $\mathbf{X} = [\mathbf{X}_1, \mathbf{X}_2, \cdots, \mathbf{X}_C]$.

2.2.1. Principle of KNCCRC

Firstly, *K*-nearest classes of the test sample $y$ are chosen from the whole training sample set $\mathbf{X}$ by computing the Euclidean distance between $y$ and training samples of class-specific $\mathbf{X}_l, l \in \{1, 2, \cdots, C\}$. Its formula is expressed as

$$d(\mathbf{X}_l, y) = \min \ \{\|\mathbf{X}_l - y\|_2\} \tag{1}$$

Through Equation (1), *K*-nearest classes of the test sample $y$ can be found. Additionally, the corresponding nearest classes are used to reconstruct the dictionary and represent the test sample, in which the reconstructed dictionary can be written as $\widehat{\mathbf{X}}_K = [\mathbf{X}_1, \mathbf{X}_2, \cdots, \mathbf{X}_K]$ ($K < C$). Then, the representation coefficient vector $\boldsymbol{\alpha}$ corresponding to the reconstructed dictionary $\widehat{\mathbf{X}}_K$ for the test sample is solved by $\ell_2$-norm regularization, i.e.,

$$\boldsymbol{\alpha} = \arg \ \min_{\boldsymbol{\alpha}^*} \|y - \widehat{\mathbf{X}}_K \boldsymbol{\alpha}^*\|_2^2 + \lambda \|\boldsymbol{\alpha}^*\|_2^2 \tag{2}$$

where $\lambda$ is a global regularization parameter, which is utilized to make trade-off between the residual part and the regularization term. The representation coefficient vector $\boldsymbol{\alpha}$ in Equation (2) can be obtained with a closed-form solution in the form of

$$\boldsymbol{\alpha} = (\widehat{\mathbf{X}}_K^T \widehat{\mathbf{X}}_K + \lambda \mathbf{I})^{-1} \widehat{\mathbf{X}}_K^T y \tag{3}$$

where $\mathbf{I}$ represents the identity matrix. The obtained coefficient vector $\boldsymbol{\alpha}$ is separated into $K$ different class-specific weight coefficient vectors according to the class labels of the training samples in the reconstructed dictionary $\widehat{\mathbf{X}}_K$, i.e., $\boldsymbol{\alpha} = [\boldsymbol{\alpha}_1^T, \boldsymbol{\alpha}_2^T, \cdots, \boldsymbol{\alpha}_C^T]^T$.

Finally, the test sample $y$ is represented by the class-specific coefficient vector $\boldsymbol{\alpha}_l$ and corresponding dictionary $\mathbf{X}_l$. Additionally, the class label of $y$ is assigned to the class with the minimum residual, which is

$$\text{class}(y) = \arg \min_{l=1,\cdots,K} \|y - \mathbf{X}_l \boldsymbol{\alpha}_l\|_2^2 \tag{4}$$

### 2.2.2. Principle of KNCCRT

The KNCCRT method introduces the Tikhonov regularization term into the KNCCRC method. In the same process as with KNCCRC, $K$-nearest classes of the test sample $y$ are selected by Equation (1) to reconstruct the dictionary $\widehat{\mathbf{X}}_K$. Then, the solution equation of the representation coefficient vector $\boldsymbol{\alpha}$ corresponding to the reconstructed dictionary $\widehat{\mathbf{X}}_K$ for the test sample $y$ can be expressed in the following form:

$$\boldsymbol{\alpha} = \arg \min_{\boldsymbol{\alpha}^*} \|y - \widehat{\mathbf{X}}_K \boldsymbol{\alpha}^*\|_2^2 + \lambda \|\widehat{\boldsymbol{\Gamma}}_K \boldsymbol{\alpha}^*\|_2^2 \tag{5}$$

where the Tikhonov regularization term $\widehat{\boldsymbol{\Gamma}}_K$ is denoted as the following form:

$$\widehat{\boldsymbol{\Gamma}}_K = \begin{bmatrix} \|y - x_1\|_2 & & 0 \\ & \ddots & \\ 0 & & \|y - x_{N_{(K)}}\|_2 \end{bmatrix} \tag{6}$$

In Equation (6), $N_{(K)}$ represents the number of training samples contained in the reconstructed dictionary $\widehat{\mathbf{X}}_K$, i.e., $\sum_{l=1}^{K} N_l = N_{(K)}$. Additionally, the closed-form solution of the coefficient vector $\boldsymbol{\alpha}$ can be derived as

$$\boldsymbol{\alpha} = (\widehat{\mathbf{X}}_K^T \widehat{\mathbf{X}}_K + \lambda \widehat{\boldsymbol{\Gamma}}_K^T \widehat{\boldsymbol{\Gamma}}_K)^{-1} \widehat{\mathbf{X}}_K^T y \tag{7}$$

Finally, the obtained coefficient vector $\boldsymbol{\alpha}$ is separated into $K$ different class-specific weight coefficient vectors, and the class label of the test sample $y$ is assigned to the class with the minimum residual according to Equation (4).

### 2.2.3. Principle of the Proposed LNNCRC Method

Firstly, $k$-nearest neighbors of the test sample $y$ are selected from the class-specific $\mathbf{X}_l$ utilizing the Euclidean distance, i.e.,

$$d(x_{l,i}, y) = \|x_{l,i} - y\|_2 \tag{8}$$

where $x_{l,i}$ represents the $i$th sample in the class-specific $\mathbf{X}_l$, and the selected $k$-nearest neighbors can be written as $\mathbf{X}_l^{(k)} = [x_{l,1}, x_{l,2}, \cdots, x_{l,k}], l \in \{1, 2, \cdots, C\}$.

The training samples contained in $\mathbf{X}_l^{(k)}$ are regarded as the local nearest neighbors of $y$ in the $l$th class. Then, the similarity between $y$ and each class-specific $\mathbf{X}_l^{(k)}$ is evaluated

by calculating the density of local nearest neighbors. The specific formula is expressed as follows:

$$\rho_l = \sum_{i=1}^{k} \exp(-\|x_{l,i} - y\|_2) \tag{9}$$

where $x_{l,i}$ is the $i$th training sample in the class-specific $\mathbf{X}_l^{(k)}$. The larger the value of $\rho_l$ is, the more similar the test sample $y$ is to the class-specific $\mathbf{X}_l^{(k)}$. The values of $\rho_l$ for all classes are sorted in ascending order, and the $K$ top-ranked class-specific $\mathbf{X}_l^{(k)}$ are selected to reconstruct the dictionary, which only contains $K$-nearest classes of the test sample $y$ and $k$-nearest training samples of $y$ in each class, denoted as $\widehat{\mathbf{X}}_K^{(k)} = [\mathbf{X}_1^{(k)}, \mathbf{X}_2^{(k)}, \cdots, \mathbf{X}_K^{(k)}]$ $(K < C)$.

Then, the reconstructed dictionary $\widehat{\mathbf{X}}_K^{(k)}$ is employed to represent the test sample $y$, and the representation coefficient vector $\boldsymbol{\alpha}$ is solved according to Equation (10), i.e.,

$$\boldsymbol{\alpha} = \arg \min_{\boldsymbol{\alpha}^*} \|y - \widehat{\mathbf{X}}_K^{(k)} \boldsymbol{\alpha}^*\|_2^2 + \lambda \|\boldsymbol{\alpha}^*\|_2^2 \tag{10}$$

Additionally, the closed-form solution of the coefficient vector $\boldsymbol{\alpha}$ in Equation (10) can be denoted as

$$\boldsymbol{\alpha} = ((\widehat{\mathbf{X}}_K^{(k)})^T \widehat{\mathbf{X}}_K^{(k)} + \lambda \mathbf{I})^{-1} (\widehat{\mathbf{X}}_K^{(k)})^T y \tag{11}$$

Finally, LNNCRC method allocates class label to the test sample $y$ in the same way as the KNNCRC and KNNCRT methods; the obtained coefficient vector $\boldsymbol{\alpha}$ is separated into $K$ different class-specific weight coefficient vectors, and the class label of $y$ is assigned to the class with the minimum residual, i.e.,

$$\text{class}(y) = \arg \min_{l=1,\cdots,K} \|y - \mathbf{X}_l^{(k)} \boldsymbol{\alpha}_l\|_2^2 \tag{12}$$

### 2.2.4. Principle of The Proposed LNNCRT Method

The LNNCRT method is the Tikhonov regularized version of LNNCRC. As the same with LNNCRC, $k$-nearest neighbors of the test sample $y$ from each class-specific $\mathbf{X}_l$ and $K$-nearest classes of the test sample $y$ are selected using Equations (8) and (9), respectively, to reconstruct the dictionary $\widehat{\mathbf{X}}_K^{(k)}$. Then, the reconstructed dictionary $\widehat{\mathbf{X}}_K^{(k)}$ is employed to represent the test sample $y$, and the representation coefficient vector $\boldsymbol{\alpha}$ is solved according to Equation (13), i.e.,

$$\boldsymbol{\alpha} = \arg \min_{\boldsymbol{\alpha}^*} \|y - \widehat{\mathbf{X}}_K^{(k)} \boldsymbol{\alpha}^*\|_2^2 + \lambda \|\widehat{\boldsymbol{\Gamma}}_K^{(k)} \boldsymbol{\alpha}^*\|_2^2 \tag{13}$$

The Tikhonov regularization term $\widehat{\boldsymbol{\Gamma}}_K^{(k)}$ is expressed as the following form:

$$\widehat{\boldsymbol{\Gamma}}_K^{(k)} = \begin{bmatrix} \|y - x_1\|_2 & & 0 \\ & \ddots & \\ 0 & & \|y - x_{\hat{N}_{(K)}}\|_2 \end{bmatrix} \tag{14}$$

where $\hat{N}_{(K)}$ is the number of training samples contained in the reconstructed dictionary $\widehat{\mathbf{X}}_K^{(k)}$. The closed-form solution of the coefficient vector $\boldsymbol{\alpha}$ can be denoted as

$$\boldsymbol{\alpha} = ((\widehat{\mathbf{X}}_K^{(k)})^T \widehat{\mathbf{X}}_K^{(k)} + \lambda (\widehat{\boldsymbol{\Gamma}}_K^{(k)})^T \widehat{\boldsymbol{\Gamma}}_K^{(k)})^{-1} (\widehat{\mathbf{X}}_K^{(k)})^T y \tag{15}$$

Finally, in the same process as LNNCRC, the obtained coefficient vector $\boldsymbol{\alpha}$ is separated into $K$ different class-specific weight coefficient vectors, and the class label of the test sample $y$ is assigned to the class with the minimum residual according to Equation (12).

## 3. Results and Discussion

### 3.1. Parameter Optimization

In this paper, the proposed LNNCRC and LNNCRT methods are compared with the CRC, CRT, NSC, NRS, KNCCRC, and KNCCRT methods for land cover classification performance under their respective optimal parameters, so as to fairly verify the effectiveness of the proposed methods. In the process of parameter optimization, 10% and 20% of the labeled samples from each class are randomly selected as the training samples and validation samples, respectively. Additionally, the overall accuracy (OA) is used to evaluate the performance of classification models for each parameter. To avoid random error and any bias, each classification model was run 10 times under each parameter and took the average value as the final result.

The global regularization parameter $\lambda$ is the main parameter affecting the classification performance of CRC, CRT, NSC, and NRS. For the KNCCRC and KNCCRT methods, both $\lambda$ and the nearest-class parameter $K$ are the main parameters that affect the classification performance. Additionally, for the proposed LNNCRC and LNNCRT methods, $\lambda$, $K$, and the local nearest neighbor parameter $k$ are the three main parameters. It can be seen that $\lambda$ is the common parameter for all of the above-mentioned methods. In the optimization process, $\lambda$ of NSC and NRS is chosen from the given intervals {3.5, 4, 4.5, 5, 5.5, 6, 6.5, 7, 7.5, 8}, and $\lambda$ of the other methods is chosen from the given intervals {$10^{-3}$, $3 \times 10^{-3}$, $5 \times 10^{-3}$, $10^{-2}$, $3 \times 10^{-2}$, $5 \times 10^{-2}$, $10^{-1}$, $3 \times 10^{-1}$, $5 \times 10^{-1}$, 1}. The parameter $K$ is chosen from 1 to 9 in intervals of 1, because there are nine land cover types in the acquired hyperspectral scene. Additionally, the parameter $k$ is chosen from the given intervals {15, 20, 25, 30, 35, 40, 45, 50, 55, 60}. Figure 2 shows the classification performance of the CRC, CRT, NSC, and NRS methods under different $\lambda$ values. Additionally, Figure 3 shows the classification performance of the KNCCRC, KNCCRT, LNNCRC, and LNNCRT methods under different parameters, in which an asterisk (*) is used to represent the position of the optimal parameters. Moreover, the surface of different colors is employed to represent the corresponding nearest-class parameter $K$ in the three-dimensional graph of LNNCRC and LNNCRT, as shown in Figure 3c,d. There is a list of the optimal parameter settings for each method in Table 2.

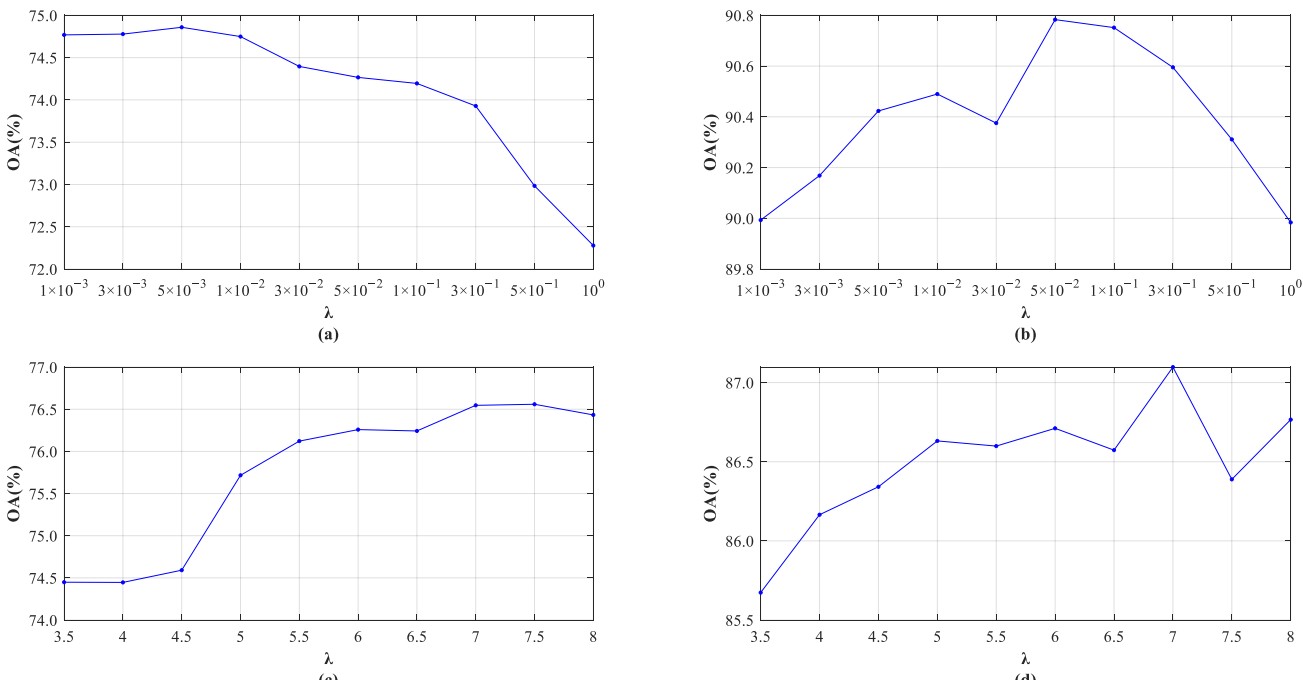

**Figure 2.** Classification performance for (**a**) CRC, (**b**) CRT, (**c**) NSC, and (**d**) NRS under different $\lambda$ values.

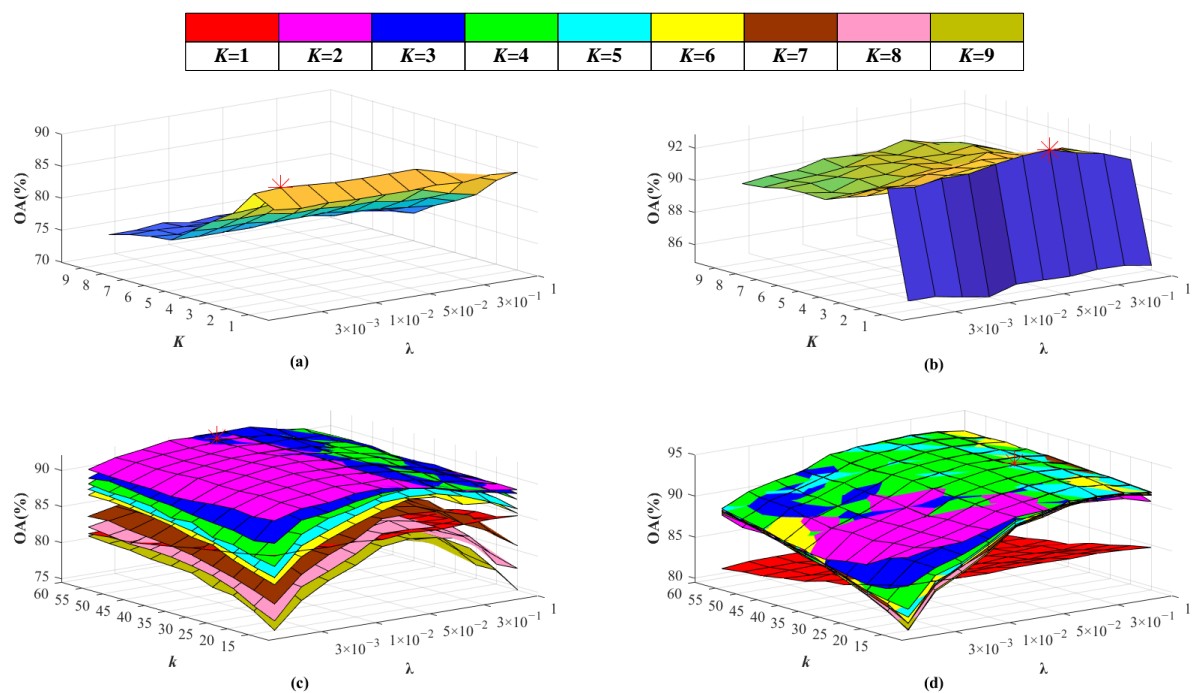

**Figure 3.** Classification performance for (**a**) KNCCRC, (**b**) KNCCRT, (**c**) LNNCRC, and (**d**) LNNCRT under different parameters.

**Table 2.** Optimal parameter settings for each method.

| Parameters | Methods | | | | | | | |
|---|---|---|---|---|---|---|---|---|
| | CRC | CRT | NSC | NRS | KNCCRC | KNCCRT | LNNCRC | LNNCRT |
| $\lambda$ | $5 \times 10^{-3}$ | $5 \times 10^{-2}$ | 7.5 | 7 | $3 \times 10^{-3}$ | $1 \times 10^{-1}$ | $3 \times 10^{-2}$ | $3 \times 10^{-1}$ |
| $K$ | No application | No application | No application | No application | 2 | 2 | 2 | 4 |
| $k$ | No application | No application | No application | No application | No application | No application | 40 | 55 |

### 3.2. Land Cover Classification Performance for Different Methods

In this section, the classification performance of the proposed LNNCRC and LNNCRT methods is compared with that of six other popular methods under the corresponding optimal parameters. As mentioned earlier, 10% and 20% of the labeled samples from each class were used as the training set and validation set, respectively, so the remaining 70% labeled samples were selected as the test set to evaluate the land cover classification performance for each method. Additionally, individual class accuracy, overall accuracy (OA), average accuracy (AA), and kappa statistic (kappa) were utilized as evaluation indicators. Similarly, to avoid random error and any bias, each classification model was run 10 times under the corresponding optimal parameters and took the average value as the final result. The classification results can be seen in Table 3 and Figure 4, in which the best classification results are presented in highlighted font in Table 3.

**Table 3.** Classification accuracy of different methods for land cover types.

| Class | CRC | CRT | NSC | NRS | KNCCRC | KNCCRT | LNNCRC | LNNCRT |
|---|---|---|---|---|---|---|---|---|
| *Asphalt* | 95.99 | 92.99 | **96.32** | 95.51 | 94.12 | 92.73 | 89.83 | 92.85 |
| *Meadows* | 98.51 | 99.46 | **99.83** | 99.55 | 98.30 | 99.24 | 97.68 | 98.32 |
| *Gravel* | 23.95 | 65.15 | 32.29 | 44.49 | 62.30 | 73.31 | 65.51 | **75.17** |
| *Trees* | 82.53 | 90.28 | 53.10 | 82.86 | 84.61 | 90.90 | **92.63** | 92.38 |
| *Painted metal sheets* | 71.75 | 98.40 | 98.25 | 99.25 | 81.32 | **99.18** | 99.11 | 99.10 |
| *Bare soil* | 23.82 | 72.96 | 34.61 | 47.80 | 65.61 | 81.20 | **88.05** | 86.25 |
| *Bitumen* | 0.00 | 64.91 | 1.36 | 33.03 | 36.94 | 77.14 | 75.55 | **82.23** |
| *Self-blocking bricks* | 47.63 | 84.21 | 54.38 | **92.13** | 80.10 | 87.90 | 88.10 | 86.60 |
| *Shadows* | 4.11 | 99.77 | 36.36 | 99.41 | 99.71 | **99.98** | 99.91 | 99.89 |
| OA (%) | 74.17 | 90.59 | 76.53 | 86.22 | 87.08 | 92.59 | 91.97 | **93.04** |
| AA (%) | 49.81 | 85.35 | 56.28 | 77.12 | 78.11 | 89.06 | 88.49 | **90.31** |
| Kappa | 0.6358 | 0.8729 | 0.6666 | 0.8103 | 0.8247 | 0.9006 | 0.8931 | **0.9071** |

Although both KNCCRC and KNCCRT methods select the *K*-nearest classes of each test sample from all the labeled training samples to represent and classify the test sample [24], they do not consider the impact of the imbalanced number of training samples from each class on the classification performance; meanwhile, the proposed LNNCRC and LNNCRT methods select the same number of nearest neighbors of each test sample from each of *K*-nearest classes to construct dictionary. Additionally, the results from Table 3 shows that LNNCRC and LNNCRT outperform KNCCRC and KNCCRT, respectively, which indicates that the proposed methods not only further eliminate the interference of irrelevant training samples and classes, but also effectively eliminate the influence of imbalanced training samples on classification performance. Especially, the proposed LNNCRT method achieves the highest OA (93.04%), AA (90.31%), and kappa (0.9071) for land cover classification. Additionally, compared with other methods, the classification noise in the land cover classification map of LNNCRT is the lowest, as shown in Figure 4i.

Furthermore, the CRT [21], NRS [20], KNCCRT [24], and LNNCRT methods are the Tikhonov regularized versions of CRC, NSC, KNCCRC, and LNNCRT, respectively. Additionally, it can be seen from Table 3 that CRT, NRS, KNCCRT, and LNNCRT achieve a better classification performance than CRC, NSC, KNCCRC, and LNNCRC, respectively, which further verifies that the Tikhonov regularization term can effectively improve the classification performance of CR models.

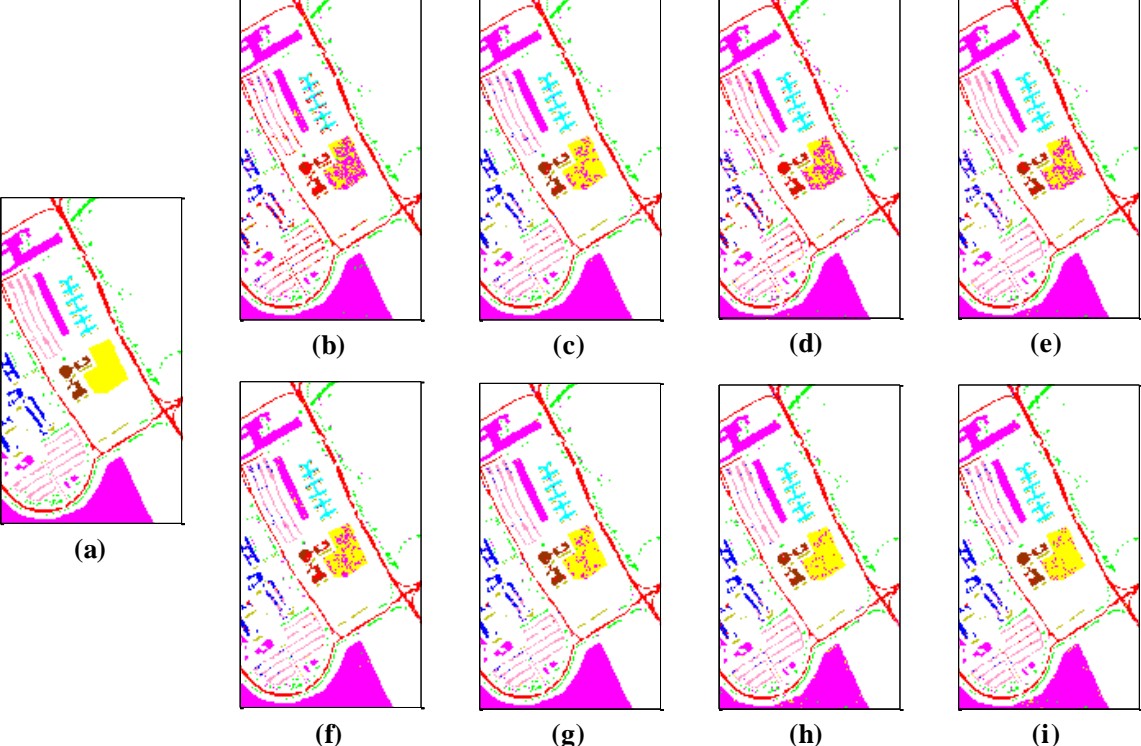

**Figure 4.** (**a**) Ground truth map and land cover classification maps generated by (**b**) CRC, (**c**) CRT, (**d**) NSC, (**e**) NRS, (**f**) KNCCRC, (**g**) KNCCRT, (**h**) LNNCRC, and (**i**) LNNCRT.

In addition, compared with other methods, the CRC, CRT, NSC, and NRS methods have a poor performance for land cover classification. The reason for this is that these methods all use training samples to construct dictionaries, which contain many training samples unrelated to test samples, thus affecting the classification performance. Especially, the CRC method possesses the worst performance for land cover classification, in which OA, AA, and kappa are only 74.17%, 49.81%, and 0.6358, respectively. Additionally, there is the most classification noise in the land cover classification map of CRC as shown in Figure 4b.

### 3.3. Comparison of Running Time

In this section, the running times of different methods for land cover classification are compared using the MATLAB R2014a software on a computer with 2.90 GHz CPU and 32 GB RAM. To avoid random error and any bias, each method runs 10 times and takes the mean running time as the final result, which is shown in Table 4.

**Table 4.** Running time of different methods for land cover classification.

| Methods | CRC | CRT | NSC | NRS | KNCCRC | KNCCRT | LNNCRC | LNNCRT |
|---|---|---|---|---|---|---|---|---|
| Running time (seconds) | $5.2323 \times 10^1$ | $3.9765 \times 10^4$ | $5.8659 \times 10^3$ | $6.0098 \times 10^3$ | $1.2621 \times 10^4$ | $1.2867 \times 10^4$ | $7.3053 \times 10^1$ | $2.8159 \times 10^2$ |

The results show that the CRT, NRS, KNCCRT, and LNNCRT methods take much more time than CRC, NSC, KNCCRC, and LNNCRC, respectively, which indicates that the Tikhonov regularization term can effectively improve the performance of CR models for land cover classification but significantly increases the running time of CR models. Compared with the running time of CRT, NRS, and KNCCRT, the proposed LNNCRT method has the shortest running time, which indicates that the proposed local nearest neighbor method can effectively reduce the computational complexity of CR models with

Tikhonov regularization. Moreover, for the CR models without Tikhonov regularization, the proposed LNNCRC method takes much less time than NSC and KNCCRC. Although LNNCRC takes slightly more time than CRC, the land cover classification performance of LNNCRC is significantly better than that of CRC. Therefore, the proposed LNNCRC and LNNCRT methods not only improve the land cover classification performance of CR models, but also effectively reduce the computational complexity of CR models.

## 4. Conclusions

In recent years, the collaborative representation (CR) classification model has been widely used in land cover classification from hyperspectral images. However, most of the existing CR models do not take the problem of sample imbalance into account, which may cause a great impact on the performance of CR models. In addition, many CR models employ the Tikhonov regularization term to improve their classification performance, which greatly increases the computational complexity of CR models. To solve the above problems, the LNNCRC and LNNCRT methods are proposed for land cover classification from hyperspectral images in this paper, in which LNNCRT is the Tikhonov regularized version of LNNCRC. The essential idea of the proposed methods is to select the same number of nearest neighbor samples from each nearest class of the test sample to construct a dictionary via the designed local nearest neighbor (LNN) method. Additionally, the conclusions are summarized as follows:

(1)   Compared with other methods, the proposed LNNCRT method achieves the best land cover classification performance, in which the OA, AA, and kappa reach 93.04%, 90.31%, and 0.9071, respectively.

(2)   LNNCRC and LNNCRT outperform KNCCRC and KNCCRT, respectively, which indicates that the proposed methods not only further exclude the interference of irrelevant training samples and classes, but also effectively eliminate the influence of imbalanced training samples, so as to improve the land cover classification performance of CR models.

(3)   LNNCRT takes much less time than CRT, NRS, and KNCCRT, and LNNCRC takes much less time than NSC and KNCCRC, which indicates that the proposed methods can effectively reduce the computational complexity of CR models.

It can be seen from the experimental results on a hyperspectral scene with nine land cover types that the proposed LNNCRC and LNNCRT methods not only improve the classification performance of CR models, but also effectively reduce the computational complexity of CR models. However, the variations in light condition, physical location and season usually lead to a shift in the spectral curves of the same ground objects [13], resulting in a nonlinear structure of the sample data. The linear representation of the proposed CR models is insufficient for representing this nonlinear structure. In addition, the proposed methods do not utilize the spatial features of hyperspectral images. In future research, a spatial–spectral weighting mechanism and kernel function will be introduced into the proposed CR models to further improve their land cover classification performance.

**Author Contributions:** Data curation, R.Y.; Methodology, R.Y.; Supervision, Q.Z. and Y.W.; Validation, B.F.; Writing—original draft, R.Y.; Writing—review and editing, R.Y. and B.F. All authors have read and agreed to the published version of the manuscript.

**Funding:** This research was supported by the Basic Research Fund of Agricultural Information Institute of CAAS (Grant No. JBYW-AII-2022-02) and the Innovation Research Fund of Agricultural Information Institute of CAAS, China (CAAS-ASTIP-2016-AII).

**Institutional Review Board Statement:** Not applicable.

**Informed Consent Statement:** Not applicable.

**Data Availability Statement:** The hyperspectral data set of Pavia University can be obtained from http://www.ehu.eus/ccwintco/index.php?title=Hyperspectral_Remote_Sensing_Scenes. And we obtained this data in 1 July 2021.

**Acknowledgments:** We acknowledge Paolo Gamba from Pavia University for providing the ROSIS hyperspectral data from Pavia University for the research of land cover classification.

**Conflicts of Interest:** The authors declare no conflict of interest.

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
