# Peer review of "Land Cover Classification from Hyperspectral Images via Local Nearest Neighbor Collaborative Representation with Tikhonov Regularization"

_land, doi:10.3390/land11050702_

Round 1
Reviewer 1 Report
I am very positive about the topic presented in the article. It is interesting and involves new research issues. The title of the article reflects this well. The abstract comprehensively explains the problematic undertaken in the article. I also positively assess the Introduction, which contains a comprehensive literature review. The methods are presented correctly.
I make minor comments on the results and discussion section. The discussion definitely lacks reference to the theses in the literature. In my opinion, this element should be supplemented to a greater extent. The conclusions are also too modest. The authors limited themselves to making a few points. In my opinion, it would have been much more justified to describe them more extensively.
Reviewer 2 Report
Review of the manuscript entitled “Land Cover Classification from Hyperspectral Images via Local 2 Nearest Neighbor Collaborative Representation with Tikhonov Regularization"
General comments:
The authors proposed a LNNCRC and LNNCRT methods to perform a land cover classification and compared the results with six other popular CR models on a hyperspectral scene with 9 land cover types. Their results revealed that the proposed LNNCRT method resulted in optimal land cover classification. The authors suggest the proposed LNNCRC and LNNCRT methods provides an added benefit of further excluding the interference of irrelevant training samples and classes and effectively eliminate the influence of imbalanced training samples to improve the classification performance of CR models and effectively reduce the influence of imbalanced training samples the computational complexity of CR models.
In general, I find the manuscript is interesting and informative but the manuscript in current form needs significant language editing. In my opinion, the manuscript is suitable for publication in land journal, after the authors have addressed the following comments and questions.
The points to consider are summarized in the table below.
a) Title: |
Overall, the title is appropriate for the paper. |
b) Abstract: |
Overall, the abstract is well-written and informative. Minor comments Line 13: insert “the” to read “… and the collaborative representation …” Line 20: insert “a” to read “…construct a dictionary …” Line 25-29: Rephrase the entire sentence |
c) Keywords: |
The authors provided five (5) keywords. It is within the journal's specified number. The keywords are appropriate to improve the article search results in the future or increase the article's visibility to a large audience. |
d) Introduction |
The introduction is well written, and the motivation of the study is relevant. However, I suggest minor language editing to improve the manuscript. Minor comments Line 35: replace “for” with “in” to read “… changes in the earth’s surface” Line 36: change “of great significance” to “significant” Line 41: split sentence to read: “… ground objects. Still, they can …” Line 46: delete “s” in “provide” Line 60: insert “a” to read “… does not need a complex ...” Line 70: insert “the” to read “… advantages of the collaborative …” Line 75: replace “original CRC for HIS classification, which was” into “the original CRC for HIS classification,” Line 78: split sentence to read “… (called pre-partitioning). In contrast, …” Line 79: space Line 88: delete “so as” Line 91: replace “imposes a great influence on” to “greatly influences” Line 105-108: rephrase the entire sentence Line 108: insert “to” to read “collaboratively to represent the …” Line 112: delete “At present,” Line 117: change “so on” to “etc.” Line 124: insert “a” to read “construct a dictionary” Line 130: replace “are” with “to” Line 133: insert “the” to read “the K-nearest …” Line 134: insert “a” to read “construct a dictionary”; split sentence and delete “in which” to read “Each nearest …” Line 136: rephrase to read “… represents and classifies the test samples” |
e) Methodology
|
The methodology is clear, transparent, and is appropriate for this study. The research procedures and techniques used are standard and reproducible. However, I will suggest authors include limitations encountered in this section. Minor comments I strongly suggest English language editing to improve the manuscript. |
f) Results, discussion and Conclusion |
The results are clear, well presented and is well complemented with tables and a figure to help in results visualization. The discussion section supports many aspects of the findings and done within the content of previous studies. Minor comments Line 348: include the version and citation to MATLAB I strongly suggest English language editing to improve the manuscript. |
g) Reference |
Most of the references are current and relevant for the study. |
h) Recommendation |
Accept with minor revision |

Reviewer 3 Report
Good for the publication in the present form
Reviewer 4 Report
The paper proposed a local nearest neighbor collaborative representation classification method to improve hyperspectral images classification quality. Then use classification performance and running time of the proposed methods as a quality checking matrix to compare the effectiveness of the proposed method. The background analysis was well organized and clearly presented. The methodology and results were clearly presented. However, the conclusion was too simple. It is suggested to summarize the reason the method was proposed, the key methodology improvements and discuss possible implications. The language needs to be double-checked before publication.
